# The Role of miRNAs and Extracellular Vesicles in Adaptation After Resistance Exercise: A Review

**DOI:** 10.3390/cimb47080583

**Published:** 2025-07-23

**Authors:** Dávid Csala, Zoltán Ádám, Márta Wilhelm

**Affiliations:** 1Institute of Sport Sciences and Physical Education, Faculty of Science, University of Pécs, Ifjúság Str. 6, 7624 Pécs, Hungary; csaladav@gamma.ttk.pte.hu; 2Doctoral School of Biology and Sport Biology, Faculty of Science, University of Pécs, Ifjúság Str. 6, 7624 Pécs, Hungary; 3Department of Pharmaceutical Biotechnology, Faculty of Pharmacy, University of Pécs, Rókus Str. 2, 7624 Pécs, Hungary

**Keywords:** resistance training, exosome, microvesicle, intercellular communication, physical activity, signaling

## Abstract

Resistance exercise can enhance or preserve muscle mass and/or strength. Modifying factors are secreted following resistance exercise. Biomarkers like cytokines and extracellular vesicles, especially small extracellular vesicles, are released into the circulation and play an important role in cell-to-cell and inter-tissue communications. There is increasing evidence that physical activity itself promotes the release of extracellular vesicles into the bloodstream, suggesting the importance of vesicles in mediating systemic adaptations following exercise. Extracellular vesicles contain proteins, nucleic acids like miRNAs, and other molecules targeting different cell types and tissues of distant organs. Therefore, extracellular vesicles and encapsulated miRNAs are fine tuners of protein synthesis and are important in the adaptation after resistance training. However, there is a lack of strong data supporting the precise mechanisms of these processes. In this literature review, we collected publications related to miRNA and extracellular vesicle profile changes induced by resistance exercise. To the best of our knowledge, the changes in human extracellular vesicle and microRNA profiles following resistance exercise have not been reviewed yet. We aimed to assess the shortcomings and difficulties characterizing this research area, to summarize the existing results to date, and to propose possible solutions that could help standardize the implementation of future investigations.

## 1. Introduction

Regular physical exercise can prevent diseases and preserve general health. Contracting skeletal muscle enhances the metabolic needs of the body, since changes influence all tissues and organs. Adaptations occur after training sessions, leading to better health status [1]. Resistance exercise (RE) is a training method in which repeated muscle contractions are applied against external load in combination with ideal nutrient intake, to enhance or preserve muscle mass and/or strength [2]. RE stimulates hypertrophy within skeletal muscle fibers, improving their contractile activity and the force production generated by the muscle. This improves functionality and strength, as well as overall health, protecting against age-related chronic diseases, and especially sarcopenia [3]. RE is not only important for ideal health status and rehabilitation, but it can increase sport performance by muscle hypertrophy and increased strength. Adaptive responses in the skeletal muscle take place in the regeneration phase after the workout and are influenced mostly by training intensity [4]. Hypertrophy in the muscle is due to the increase of contractile proteins, enhancing force and performance [5]. Functional and structural adaptations underlying muscle strength are extremely diverse. Both in young and elderly populations, RE causes neural and neuromuscular changes, in addition to changes in connective tissue and vascular adaptations in the muscle as well [6]. RE modifying factors are secreted by muscle fibers, playing an important role in the regulation of neuromuscular and vascular plasticity [7]. The fact that systemic factors secreted by different cell types during physical activity (PA) are released into the circulation has attracted the attention of researchers, because of the vast number of benefits in the body [1]. Recently, it was discovered that biomarkers like cytokines and extracellular vesicles (EVs), especially small EVs, are released into the circulation and serve in cell-to-cell and inter-tissue communication [8,9]. EVs contain proteins, nucleic acids like microRNAs (miRNAs), and other molecules targeting different cell types and tissues of distant organs. Therefore, these nanoparticles and encapsulated miRNAs are the fine tuners of protein synthesis and play an important role in adaptation after RE. Nonetheless, the molecular sensors and downstream signals of RE, leading to phenotypic differences, are not fully understood [10]. The endocrine system plays a crucial role in supporting the overall functioning of the body, particularly in response to physical stimuli such as RE. However, hormonal effects cannot be interpreted in isolation, as they are closely intertwined with the functions of other physiological systems. Acute hormonal responses during RE, interactions with receptors and binding proteins, the availability of receptors at the cellular level, and the secretion patterns observed in the days following exercise all contribute, at least in part, to the regulation of recovery processes [11]. In recent years, increasing attention has been directed toward the role of miRNAs and EVs in exercise-induced adaptations. Evidence suggests that miRNAs may influence these adaptive responses either locally or by being transported via EVs to target tissues or organs. This is largely attributed to the capacity of EVs to effectively deliver molecular cargo, including nucleic acids, and to traverse biological barriers such as the blood–brain barrier [12,13]. These findings suggest that EVs may serve a complementary regulatory role alongside the endocrine system in fine-tuning the physiological adaptations following resistance exercise. In several recent publications we found summaries of the effect of PA on miRNA [14] and EV profile [15,16] changes. However, we believe that this is the first review focusing only on the effects of RE in changing EV and miRNA profiles. The cited studies introduce regulating functions of nanoparticles and molecules after acute and chronic RE, in which acute and chronic RE refer to short- and long-term resistance exercise, respectively. In this summary, publications related to miRNA and EV profile changes induced by RE are collected. We wanted to assess the shortcomings and difficulties characterizing this research area, and possible solutions that could help standardize the implementation of future investigations.

## 2. Materials and Methods

To find publications related to the topic, we used the PubMed search engine, searching up to 3 June 2025 with the following terms: “resistance training mirna” (244 results), “resistance exercise mirna” (247 results), “resistance training extracellular vesicles” (58 results found), “resistance exercise extracellular vesicles” (67 results). In addition, other relevant publications related to the topic were also included in our research. The prime focus was on human studies; most studies conducted in various model organisms were excluded. Finally, 49 human studies were included which investigate and measure the effects of RE on miRNA profile. Due to the diversity of publications on the topic and the novelty of the field, it was not possible to conduct a systematic review.

## 3. Extracellular Vesicles

EVs, secreted by all cells, are particles bound by a lipid bilayer and are expressed in the extracellular environment by various cell types (Figure 1). Small EVs released by living cells can be separated into exosomes (50–150 nm) and microvesicles (50–1000 nm), but their classification into subgroups is complicated [12]. It seems appropriate to divide EVs into small (<200 nm) and large groups, although it is difficult to distinguish between them solely by size, due to overlap [17]. Exosomes are released by cells in case of multivesicular bodies (MVBs) fusing with the plasma membrane [18], while microvesicles are formed through the outward budding and fission of the membrane [19]. It is difficult to isolate a specific subunit of EVs, as there are only a few marker proteins for specification, hence the International Society for Extracellular Vesicles proposes the term “extracellular vesicle” [17]. In cases where the biogenesis of the vesicles cannot be determined, it is recommended to use the terms small EVs (less than 200 nm in diameter) or large EVs (greater than 200 nm in diameter) [20]. EVs are important players in cell-to-cell and inter-tissue communication [12], having surface molecules which enable them to target specific cells [21] and deliver molecules with signal transmission capabilities, such as proteins, metabolites, and nucleic acids (e.g., miRNAs). Once attached to the recipient cell, EVs influence biological processes by inducing cell signaling at the plasma membrane or transporting the delivered molecules into the cell [22]. Blood-derived EVs contain miRNA in huge abundance [23]. After precipitation isolation with the New Generation Sequencing (NGS) method, the isolated vesicles contained 80% miRNAs [24]. These short nucleic acids can modify gene expression of target cells by base-pairing with mRNA, thus silencing a given gene, making EVs particularly interesting from a functional point of view [25]. After determining surface protein markers and analyzing their content following stress, it would be possible to determine the tissues and cells responding to a specific stressor, thus predicting the functional role of EVs [24]. The fact that EVs are not only “garbage bags”, but they also trigger and mediate different signaling pathways between cells, motivates investigators to study the roles of EVs in various biological processes [26,27]. The physiological states of host cells are mirrored by the cargoes of EVs, suggesting that they might be used as disease or exercise biomarkers and considered as mediators of disease progression and training adaptation [28]. Furthermore, since EVs are naturally available and present in biofluids, and are easily engineered, there have been several studies using EVs as drug delivery vehicles to cure diseases [29]. Although EVs are generally capable of entering a wide range of cell types, recent studies have shown that exosomes can be non-selectively incorporated into various types of recipient cells [30]. Nevertheless, several lines of evidence suggest that a certain degree of cell-specific targeting also exists [31,32,33,34]. This targeting is primarily thought to rely on specific interactions between ligands present on the surface of EVs and receptors located on the plasma membrane of recipient cells, which may determine the binding affinity and internalization of EVs by particular cell types [13,35]. Moreover, distinct exosome subtypes may target different recipient cells due to the presence of amyloid precursor proteins or other surface molecules [36]. Therefore, although the cargo of EVs can be delivered to various cell types, targeting itself may still exhibit a certain level of cell-type specificity [13]. It is exciting to imagine the use of artificially engineered EVs in the future in sports science fields, with the potential for therapeutic application in chronic conditions linked to physical inactivity, to maximize training adaptation, boost recovery and help rehabilitation from injuries. Protein-bound miRNAs and EVs expressed as a result of RE are shown in Figure 1.

## 4. miRNAs

MiRNAs (miRs) are an evolutionarily conserved class of gene expression regulators, primarily exerting their regulatory function in the post-transcriptional phase of protein synthesis [37,38]. miRNAs are single-stranded non-coding RNA molecules, approximately 22 nucleotides long, found in plants, animals, and some viruses. Inhibition of gene expression occurs by silencing the mRNA molecule with miRNA binding to the mRNA, inhibiting translation [39]. This can happen three ways: (1), the miRNA molecule cuts the mRNA strand in half; (2), the miRNA shortens the poly(A) tail of the mRNA molecule, destabilizing it; or (3), translation through the ribosome is inhibited [40]. However, the picture is even more complicated, because one miRNA can interact with several genes, and several miRNAs can establish a connection with one gene [41]. Therefore, future research into miRNAs is particularly important since these molecules regulate at least 60 percent of the mammalian genome [42].

The first step of miRNA biogenesis happens in the nucleus, where the RNA polymerase II enzyme creates the first copy of a well-defined section of DNA [43]. This copy is the “primary” miRNA (pri-miRNA), having a guanosine cap at the 5′ end of the molecule and a poly(A) tail at the 3′ end [44]. Both have the same function: protecting the molecule against various enzymes in the body, thus hindering the elimination of the molecule. While pri-miRNA is still in the nucleus, it is recognized by two proteins, Drosha and Pasha, which cut the guanosine cap and poly(A) tail off, creating the miRNA precursor: pre-miRNA [45]. The pre-miRNA “hairpin” (the shape of the molecule resembles a hairpin) enters the cytoplasm with the aid of Exportin 5 protein [46]. In the cytoplasm, the pre-miRNA encounters the molecule Dicer cutting the loop off at the end of the double strand, forming a double-stranded pre-miRNA duplex, approximately 22 nucleotides in length [47]. One of the two strands unites with the Argonaute protein (AGO), forming the silencing complex and the mature miRNA, which is important in translation inhibition [48]. MiRNAs function as guides for AGOs, enabling the recognition of partially complementary sequences on target mRNAs and thereby facilitating subsequent gene silencing [49,50]. Furthermore, AGOs interact with the GW182 protein family (TNRC6 proteins), which mediate the recruitment of AGO complexes to target RNAs [50]. The co-localization of AGO2 and GW182 proteins to endosomes and MVBs implies that miRNA sorting into EVs may be orchestrated by the RISC complex [51,52]. Based on the Vesiclepedia database, several proteins relevant to the AGO protein network (e.g., AGO1, DDX20, DDX6, FMR1, GEMIN4, DICER1, TNRC6A, TARBP2, TNRC6C, TNRC6B, XPO5, DROSHA, MOV10, DHX9, PRKRA) have been identified in human EVs, suggesting that EVs may carry complex protein assemblies involved in RNA interference, thereby enabling the containing miRNAs to exert targeted regulation of gene expression in recipient cells.

### 4.1. Muscle-Specific miRNAs

miRNAs are particularly important in skeletal muscle myogenesis through gene regulation. This function was discovered during the deactivation of the Dicer molecule in mice, resulting in the death of the animal [53]. In addition to the endocrine effects of skeletal muscle-derived EV-encapsulated circulating miRNAs, skeletal muscle fibers regulate local processes by secreting miRNAs, fine-tuning myogenic differentiation and angiogenesis [54]. Different tissues in the body secrete tissue-specific miRNAs [55]; those exclusively or primarily coded in striated muscle are consequently called myomiRs [56]. The group currently includes eight miRNAs: miR-1, -133a, -133b, -206, -208a, -208b, -486, -499 [57,58,59]. MyomiR secretion occurs in both skeletal and heart muscle. miR-206 is exclusively expressed in the skeletal muscle, while miR-208a is cardiac muscle-specific. MyomiRs, although their primary function is related to muscle, are not only secreted in striated muscle, but also found in other tissues [60]. After the discovery of myokines, skeletal muscle can also be interpreted as an endocrine organ, since factors secreted by this tissue are relevant in the regulation of metabolic processes, after acute stress, to ensure optimal energy supply and proper muscle function [7]. It seems imaginable that miRNA-based paracrine cross-communication might play a role in skeletal muscle adaptation following exercise [7]. Only a few studies have been published in relation to RE; however, EV-encapsulated miRNAs expressed by skeletal muscle represent a promising research area in the field of endocrine, paracrine, and autocrine communication. To understand the mechanism of RE, it is necessary to consider all the factors expressed as a result of such a load. A part of these factors are miRNA-loaded EVs.

### 4.2. Circulating miRNAs

miRNAs can be detected in various human fluids, such as serum, plasma, saliva, sweat, urine, milk, and cerebrospinal fluid, offering opportunities to study them [61]. miRNAs appearing in the blood circulation are called circulating miRNAs (ci-miRNAs). In the bloodstream, ci-miRNAs are transported to the target cells with the aid of EVs (microvesicles, exosomes), proteins (Argonaute), or high-density lipoproteins (HDL), and might affect the translation of complementary mRNA [14]. There is increasing evidence that PA itself promotes the release of exosomes from skeletal muscle and other tissues into the bloodstream, suggesting that EVs are important in mediating systemic adaptations following exercise [9]. MiRNA transport is a well-supported mechanism between cells, as well as the functional regulation of gene expression of target cells as part of the communication between them; however, the exact mechanism is still unclear [62]. Findings demonstrated that after administration of miRNA-containing EVs derived from muscle, following high-intensity interval training, glucose tolerance of untrained mice improved [63]. MiRNAs delivered in EVs are taken up by target cells, which might influence the gene expression of the target [64]; however, it is not entirely clear if free miRNAs are able to attain the same function [65]. Based on the published studies so far, PA as a stressor plays a significant role in changing the miRNA profile, therefore miRNAs can also serve as PA biomarkers. A better understanding of adaptations following physical exertion at the molecular level and the effect of PA could help us to understand the exact function of miRNAs.

Most published studies so far have examined the miRNA profile changes after acute, aerobic exercise (AER). Nair et al. [66] compared the miRNA profile following acute endurance exercise in trained and untrained elderly individuals and found that miRNAs were expressed differently at rest and after exercise. The IGF-1 signaling channel was activated in the trained population, while inhibited in the untrained group, suggesting that fitness status can counteract age-related anabolic resistance through changes of small EVs. In the study, only one method was used to identify nanoparticles, hence one cannot be certain that the examined particles were exosomes. In such cases, the “small EV” term is more appropriate [17,20], since in the absence of marker proteins, determining the biogenesis of the isolated particles is not possible. Therefore, Minimal Information for Studies of Extracellular Vesicles (MISEV 2018 and 2023) recommends at least three procedures for the characterization of EVs: protein marker testing, techniques providing images of single EVs, and single particle analysis techniques [17,20]. Similar to endurance training, RE also increases the amount of ci-miRNA, depending on the type of exercise modality [67]. Previous studies failed to detect significant difference in the effect of RE training, probably because only a few selected miRNAs were examined [68,69,70].

## 5. Ci-miRNAs After Acute RE

Table 1 summarizes the acute, within-group changes in ci-miRNA and muscle biopsy miRNA levels in response to RE, along with key influencing factors characteristic of each study. These studies did not examine miRNA profiles isolated specifically from EVs, but rather assessed the total miRNA profile obtained from serum or plasma. As such, the data include miRNAs associated with proteins, lipoproteins, and EVs. Nevertheless, these investigations provide valuable insights into the overall circulating miRNA landscape and may offer indicative information regarding EV-associated miRNAs as well. In a previous study, subjects performed RE, and significant positive change of ci-miR-133 after acute exercise was demonstrated [71]. No immediate miRNA profile changes were detected afterwards; however, a day later significant increase was measured in ci-miR-149, while three days after exercise the amount of -146 and -221 significantly decreased. In that study, ci-miR-21 correlated with adrenaline and norepinephrine, while ci-miR-222 correlated with IGF-1 and testosterone [72]. In contrast to the previous studies, Margolis et al. [73] compared young and elderly men after RE. MiRNA profiles from 90 serum samples were analyzed with qRT-PCR. Fasting ci-miRNA expression of miR-19b-3p, miR-206, and miR-486 was significantly different between groups. Using Ingenuity Pathway Analysis, ci-miRNA and mRNA interaction was investigated, supported by the phosphorylation status of p-Akt^Ser473^ and p-S6K1^Thr389^ and expression of miR-19a-3p, -19b-3p, -20a-5p, -26b-5p, -143-3p, and -195-5p; the conclusion reached was that dysregulation of ci-miRNAs as a result of aging is not only a predictive marker, but also helps understanding of the molecular changes underlying skeletal muscle decline [73]. Cui and colleagues [67] compared ci-miRNA profile differences in muscle strength endurance, muscle hypertrophy (MH) and maximal strength-specific RE. In the strength endurance group, two c-miRNAs (miR-208b and -532) were observed to change in terms of expression, in the MH group six ci-miRNAs (miR-133a, -133b, -206, -181a, -21 and -221) did so, and two ci-miRNAs (miR-133a, -133b) changed significantly in the maximal strength group. Based on these data, similar types of training with different modality and intensity distinctly affect the expression of miRNAs. MiR-532 negatively correlated with IGF-1, but positively correlated with interleukin-10. MiR-133a negatively correlated with cortisol, while a positive correlation was observed with testosterone [67]. Mir-199-3p increases myogenic differentiation and muscle regeneration. This miRNA induced MH and reduced muscle strength loss in aged mice [74]. Vogel and colleagues [75] examined the effect of acute BFR RE in young men and women, comparing the effect of low-intensity blood flow restriction (LI-BFR) on the miRNA profile to low- (LI) and high-intensity (HI) RE without BFR. LI-BFR and HI significantly increased lactate levels, correlating with miRNA-143-3p, important in arteriogenesis. Thus, they concluded that BFR might function as an external stimulus in arteriogenesis [75]. Based on these findings, miRNAs and traditional biomarkers together have great potential for monitoring adaptations following RE. In studies examining healthy young men subjected to acute RE, based on data gathered from plasma using RT-PCR [67,71,76], diverse profiles of miRNAs were observed due to variations in fitness levels, the executed exercise program, and the timing of sample collection. In all three studies, miR-133a exhibited significant but differing changes, contrasting with previous data obtained from serum in studies of similar populations [72,73,77]. However, data derived from plasma drew attention to the potential role of miR-133a, which might play a part in the adaptation following acute RE in young, healthy men. The role of miRNAs appearing in higher numbers after RE adaptation is not fully understood, since there was a significant difference between measurable miRNA profiles in the different studies. Previous investigations have used different methods in molecular profiling, reaching different results. Based on findings, one cannot ascertain when and which miRNAs are expressed after a certain load. It is advisable to investigate the miRNA profile using robust and comprehensive techniques such as NGS, array cards, or NanoString technology. The results should then be validated using qRT-PCR. Furthermore, establishing a standardized sample loading and collection protocol reflecting the fitness level of the study population would be essential for ensuring reproducibility and comparability across studies.

It is important to note that the studies included in this section examined the total ci-miRNA profile in plasma or serum, which comprises miRNAs bound to proteins (e.g., AGO2) or associated with lipoproteins, as well as those encapsulated within EVs. Since these molecules may enter the circulation through non-specific pathways rather than via targeted cell-to-cell communication, resulting data may not accurately reflect the biological processes directly related to exercise-induced adaptations. This can compromise the interpretation and reliability of the conclusions drawn. Therefore, future researchers, particularly those aiming to elucidate adaptive mechanisms triggered by RE, should primarily focus on miRNAs specifically transported by EVs.

## 6. The Relationship Between Acute RE and the miRNA Profile from Muscle Biopsy

Data of miRNA expression changes measured from muscle biopsies following acute resistance exercise are presented in Table 1. Muscle biopsies of powerlifters and untrained controls were compared at rest, finding 12 miRNAs differentially expressed [92]. Four myomiRs were elevated (miR-486, -499a, -133a, -1), and one decreased (miR-206) in healthy controls. MiR-15a, -16, -451a, -30b, -23a and -23b were elevated, while miR-126 had lower expression in powerlifters than in controls. Ci-miRNAs expressed by skeletal muscle were also investigated, after high-intensity RE in young men. This phenomenon might be a key factor in inter-tissue communications [76]. Muscle biopsies were used for miRNA detection and six miRNAs (miR-23a, -133a, 146a, -206, -378 and -486) changed significantly compared to the pre-exercise state. Among ci-miRNAs, only ci-miR-133-a and -149 changed significantly and an overlap was detected only in miR-133a, showing that miRNAs in circulation do not necessarily reflect miRNA changes in the skeletal muscle. Since they did not examine EV miRNAs, but total ci-miRNAs, there is a possibility that measuring nanoparticles would mirror changes better between circulation and tissues [76]. A recent study investigating the effects of RE on miR-1 levels found that, following exercise, miR-1 expression decreased in skeletal muscle tissue, while simultaneously increasing in the circulation, encapsulated within EVs. These EVs were shown to reach adipose tissue, where they influenced lipolytic activity [93], suggesting that RE may induce opposing miRNA expression patterns in skeletal muscle and circulation, both of which may contribute to the beneficial systemic adaptations associated with exercise. To delineate the role of circulating miRNAs in the adaptation following RE, it would be worth measuring not only changes in miRNA concentration of the recipient cell, but also mRNAs and proteins playing a role in the downstream signaling pathway. Torma et al. [87] investigated the acute effect of RE on some signal transduction-related mRNAs, miRNAs and proteins, which are important in protein synthesis, angiogenesis, mitochondrial biogenesis, antioxidant system and MH. In the study, young men performed RE; during the rest period between sets, BFR was applied on one leg, while the other leg served as a control. Muscle biopsies of subjects before and two hours post-exercise were analyzed, finding that the level of miR-206 from the BFR leg was significantly reduced and correlated with the increase of Pax7 mRNA, compared to controls. In myogenesis, the Pax7 protein is important, regulating muscle precursor cell proliferation [87]. Due to a similar effect, but lower load, BFR might be more useful for individuals with limitations, e.g., people with injuries, cardiovascular disease, and older individuals [94]. In an experiment with rodents, Koltai et al. [95] demonstrated that miRNAs are important in the regulation of MH (miR-1 and -133 decreased significantly), and this negatively correlated with skeletal muscle mass. Sullivan and colleagues [96] investigated the activation of the IGF-1/Akt/mTOR pathway, critical in maintaining hypertrophy and skeletal muscle mass. Vastus lateralis biopsies of sedentary lean and obese subjects at rest, 15 min and 3 h after acute RE were collected, finding that the expression of miR-206, a post-transcriptional inhibitor of IGF-1 expression, was higher in obese subjects compared to lean and was linked with lower IGF-1 mRNA. Examining healthy young and elderly subjects after acute RE showed 26 miRNAs differentially expressed [80]. Following this research, Russell et al. [83] found that miR-320a and -483-5p were significantly higher in the older population, which directly target SRF or MRTF-A.

In studies investigating healthy young men subjected to acute lower limb RE based on data obtained from biopsies (vastus lateralis) using RT-PCR [76,86,89], different miRNA profiles were observed due to variations in fitness levels and the timing of sample collection. In all three studies, miR-133a significantly increased following acute stress [76,86,89]; this is consistent with previously mentioned data obtained from plasma in other studies. MiR-133a plays a role in myogenesis, cell fate determination, and muscle regeneration [41], thus supporting adaptation following RE. It would be useful to investigate the function of circulating EV miRNA cargos in skeletal muscle development. To better understand cell-to-cell communication following RE, it would be useful to collect not only blood, but also muscle and/or other tissue samples at multiple time points after an acute bout of exercise, in order to analyze both the levels of EV-derived miRNAs and their corresponding mRNA targets within the recipient tissues. Such analyses could provide valuable insights into the tissue-specific regulatory roles of ci-miRNAs in exercise-induced adaptations.

## 7. The miRNA Profile After RE with Protein Consumption

MiRNAs can be seen as buffers of molecular changes in the body, regulating the formation of an ideal amount of protein. Therefore, the timing of nutrient consumption after exercise cannot be fully interpreted without examining miRNA profile changes. The effect of protein consumption after acute RE was investigated in elderly men, divided into three groups, based on the amount of protein consumed: placebo, 20, or 40 g of whey protein [85], presented in Table 1. Muscle biopsies of the vastus lateralis before, 2 and 4 h post-exercise were studied, finding that miR-16-5p changed significantly in all groups, while miR-15a and -499a increased only in the placebo group 4 h after exercise, whereas miR-451 significantly increased in the 40 g protein-consuming group. The relationship between miRNAs and some proteins is important in muscle protein synthesis, and in the functioning of the Akt-mTOR channel [97], which is important in anabolism. Four hours post-exercise, p-P70S6K^Thr389^ and p-Akt^Ser473^ correlated with the expression of miR-208a, -499a and -206, regardless of the amount of protein consumed, implying that following acute stress, miRNAs are important in phosphorylation processes during regeneration [85]. A positive correlation appears between the p-Akt^Ser473^ protein and miRNA profile when measuring the expression of miRNAs [73], raising further questions regarding the role of miRNAs in the functioning of the Akt-mTOR channel. Researchers could have obtained even more accurate data by not uniformly determining the amount of nutrients consumed. Considering the body weight of individuals is a crucial factor in the efficiency of metabolism. Based on the results presented, one might compile a guideline for further research investigating the complexity of miRNA functions and the most important skeletal muscle proteins. Two studies [98,99] investigated the effects of chronic RE combined with protein intake on the miRNA profile, and are presented in Table 2. Based on the limited number of studies published to date, investigating the effects of protein intake combined with RE to modify the miRNA profile represents a promising and important direction for future research.

## 8. Studies on miRNA Profile in Response to Chronic RE

The majority of studies examining the miRNA profile in serum or plasma in response to chronic RE have focused on the elderly population [99,104,106,107,109,110,111,113,114], with only one study addressing young males [105], presented in Table 2. The listed studies examined miRNA profiles in a mixed population of males and females, often in the context of some disease or with additional intervention. Due to highly diverse research approaches so far, synthesizing the results and drawing general conclusions has been extremely challenging. Studies examining the miRNA profile in muscle biopsies of young men in response to chronic RE [81,98,100,112] have shown completely different miRNA profiles due to diverse research approaches. Chronic RE has been shown to decrease miR-1 levels in skeletal muscle tissue, based on muscle biopsy analyses [101,112], consistent with the role of miR-1 as a negative regulator of hypertrophy [115]. In contrast, other studies have reported an increase in ci-miR-1 levels following chronic training [106,113]. A similar apparent contradiction was observed in previous studies investigating acute exercise responses, raising the question of whether the acute changes in miRNA expression across different tissues may be reflected in chronic adaptations as well, suggesting a potential link between tissue-specific and circulating miRNA profiles over time. Although these particular studies did not isolate EVs, other investigations examining EV-associated miRNAs in response to acute RE have reported comparable trends [93,116]. A similar pattern was observed for miR-133, with a decrease in muscle tissue [112] and an increase in circulation [106]. However, the acute responses of this miRNA are less consistent, making it more difficult to draw clear conclusions regarding its regulation.

## 9. The Effect of RE on Extracellular Vesicle Profile Changes

Muscle contractions of PA expose muscle cells to mechanical and metabolic stress [117]; however, fat tissue, bone marrow, and the brain are also affected by the load [118]. Although it is still not entirely clear which mechanism may serve the indirect load effect, hormones, cytokines, or miRNAs transported in EVs might all be important players. This supports the idea that stress, promoting protein synthesis and releasing molecules such as miRNAs carried in EVs, reaching distant organs and interacting with many other cells, would influence their physiological function [119]. Several studies have examined free and EV-packaged ci-miRNAs released during PA. Most of these studies characterized myomiRs (for example: miR-1, 206, 133a and 133b) expressed by muscle cells under different exercise models [120]. miRNA-containing EVs are released following PA, but their exact physiological impacts are still unknown. Recent studies suggest that muscle immensely participates in the expression of EVs under physical stress [9,121]. Furthermore, EVs released following PA, similarly to hormones and cytokines, might mediate gene expression processes of the target cell [64]. Therefore, in Table 3 we summarized the changes in EV-associated ci-miRNA profiles induced by RE, as these alterations may help to better understand the targeted intercellular communication occurring during the adaptive response to RE.

A previous study investigated the level of miRNAs transported in EVs, expressed after RE. Plasma was collected from healthy young male subjects before and two hours after exercise and a significant increase in mir-146a and mir-206 transported in EVs was demonstrated when comparing the before and after loads [65]. miR-206, together with myomiRs such as miR-1 and miR-133b, are skeletal muscle-specific or highly secreted by muscle tissue, thus important in muscle development and differentiation [5]. Therefore, Annibalini et al. [65] suggested that skeletal muscle also participates in the expression of circulating EVs, since the post-RE EV-packaged miR-206 amount increased [121]. After stress, many miRNAs play a major role in inflammatory processes [126]. Among them, miR-146a is crucial for the development of the innate and adaptive immune responses, negatively regulating TLR4 signaling and NF-κB-induced pro-inflammatory cytokine and chemokine expression [127]. The increase in EV-packaged miR-146a may be important in the inflammatory processes resulting from RE [126]. Hence, circulating EVs might be important markers of muscle function and adaptation after exercise [9,121]. Changes of EV profiles after RE in men and women were also investigated [128]. Sampling (plasma) took place immediately before and after exercise. The exosome marker protein CD63 and the EV concentration showed a significant increase in both sexes, but to a different extent. In men, the average size of EVs decreased, which might be the consequence of an increase in EVs in the smaller size range. VAMP3 (microvesicle marker protein) and EV concentration showed a different, but significant increase in both sexes, suggesting that EV profile might be sex-specific in the adaptation following acute RE [128]. Changes in the miRNA profile of EVs following acute BFR RE among young men was probed, sampling plasma immediately before and 1 h post-exercise [24]. The miRNA content of EVs was determined by NGS, providing the possibility of monitoring the entire miRNA profile. No significant difference was observed in myomiRs, consistent with the results of a similar publication [67]. Based on these results, it is suggested that hypoxic environment, shear force and muscle contractions have a significant effect on blood cells and bone marrow. This could be particularly interesting, since a significant portion of EVs measured in plasma are derived from platelets, red blood cells and monocytes [129]. Considering the entire EV pool, only 1–5% of them are muscle-specific [24]. Researchers measured a significant difference in 12 miRNAs compared to the pre-load state, mainly derived from blood cells and bone marrow. Following the functional enrichment analysis of the 12 miRNAs, it was concluded that these miRNAs play a role in the regulation of cell cycle and cell growth, and may also be involved in the mTOR pathway, which is an important regulator of adaptation following PA. [24]. In future studies, one should collect blood in a citrate-containing tube (ACD-A: 3% citrate content, ACD-B: 2% citrate content) [130] followed by the platelet-free plasma (PFP) centrifugation procedure [131], preventing ex vivo vesicle release from platelets. The effects of an 8-week RE program were investigated in 38 elderly subjects, males and females (28 trained and 10 controls) [111]. The levels of miR-146a-5p (total miRNA extraction), cell free DNA, exosome markers (Flot-1, CD9, and CD81), as well as exosome-carried proteins (CD14 and VDAC1) isolated from plasma did not change after the exercise program. However, CD63 was attenuated in the trained group compared to controls. Vesicles were isolated using DUC and filtration and detected with the Western blot (WB) technique. In the research of Xhuti and colleagues [123], elderly subjects performed home-based RE for 12 weeks, compared to young subjects not participating in the exercise program. Blood samples and muscle biopsies were taken from the elderly pre- and post-exercise. After PFP isolation, small EVs were isolated using size exclusion chromatography (SEC) and ultracentrifugation (UC). The isolated vesicles were detected by immunoblotting, nanoparticle tracking analysis (NTA) and transmission electron microscopy (TEM). Total RNA was isolated from vesicles, isolated from PFP and from skeletal muscle, after which the selected miRNAs were analyzed using RT-PCR [111]. The expression of miR-23a and -27a increased significantly after 12 weeks of exercise, suggesting functions of these miRNAs in reducing skeletal muscle atrophy [132,133]. Additionally, the expression of miRNA-199a almost significantly increased, while miR-146a and -92a were significantly increased [123]. The miRNA profile of skeletal muscle showed no major difference after exercise; however, miR-1, -206, -181a, -92a, -422a, -23a, -23b, -34a differed notably, when comparing the young and old population. MiR-181a might be important in mitochondrial dynamics [134], while miR-422 is important in atrophy and MH [135]. Data show that RE can partially normalize the expression of ELV-miRNAs, which is important in maintaining health [123]. The effect of acute RE in obese and lean sedentary men and women was investigated in terms of molecular changes supporting muscle capillarization [136]. Components of the MVB biogenesis, miRNA processing, and small EV release pathways are responsive to acute AER and concurrent (aerobic and resistance) exercise [137]. The effect of aging was studied in terms of the expression of angiogenic growth factors, miRNA, MVB biogenesis and EV release influenced by rest and RE in human muscle. Participants completed three sets of single-leg RE. Baseline muscle biopsy was obtained from the non-exercised leg immediately before exercise, followed by biopsies in the RE leg at 15 min and 3 h post-exercise. The expression of mRNAs and miRNAs involved in the biogenesis of EV and miR-126, -130a, 133a, -206, -503 were measured by qRT-PCR, while the change in VEGF was determined by enzyme-linked immunosorbent assay (ELISA). Acute RE increased CD-63 mRNA at 3 h. Angiogenesis-associated miRNAs, miR-130a and -503, did not change in response to acute RE. There were no effects of obesity or acute RE on the EV release (Rab11, Rab27A, Rab27B, Rab35 or syntaxin1A mRNA) pathway. Acute RE reduced the exosome surface protein Alix, which is consistent with the release of small EVs in response to exercise [138]. TSG-101 level increased 3 h post-exercise [136]. EVs as biomarkers were investigated after eccentric RE [139] of physically active, healthy young and elderly subjects performing eccentric unilateral leg presses (seven sets of 10 reps). EVs were isolated from the plasma taken by UC, and particles were identified using NTA, WB (CD63, Flotillin-1) and TEM. The mode and concentration of EVs did not change after exercise [139]. In a study of young, Olympic-level resistance-trained male athletes, it was demonstrated that the levels of miR-16-5p, miR-19a-3p, and miR-451a within EVs were significantly lower compared to those in a sedentary control group. Thus, RE appears to exert long-term effects on the levels of microRNAs involved in inter-tissue communication, potentially reflecting systemic adaptive processes within the organism [140]. In another study conducted on older individuals, a set of miRNAs derived from skeletal muscle and serum EVs were identified to be associated with increases in thigh lean mass [141]. Previous studies have shown that, in response to mechanical overload in mice, skeletal muscle communicates with adipose tissue by releasing muscle-specific miR-1 through EVs, thereby influencing key cell signaling pathways within the adipose tissue, because of enhanced catecholamine sensitivity [116].

A recent study from the same research group has confirmed and extended previous findings. Following RE, EVs derived from skeletal muscle, carrying the muscle-specific miR-1, may play a significant role in EV-mediated inter-tissue communication between skeletal muscle and adipose tissue. In addition to classical statistical analyses, the application of machine learning models provided deeper insights and revealed novel associations. Transcriptomic sequencing of adipose tissue indicated that miR-1 may regulate lipolysis through the inhibition of CAV2 and TRIM6. Furthermore, in vitro results demonstrated that miR-1 can potentiate epinephrine-induced lipolytic activity [93]. These findings indicate that, in addition to the more extensively studied myokines, skeletal muscle can also influence lipolysis through EVs, suggesting that RE may induce active lipid mobilization via EV-mediated mechanisms following mechanical loading. Based on these findings, it appears that the reduction in miR-1 levels in skeletal muscle following RE contributes not only to the establishment of a local anabolic environment, but may also promote a catabolic state in adipose tissues located elsewhere in the body. This further underscores the role of skeletal muscle as an endocrine organ involved in systemic metabolic regulation. Future research should consider investigating the effects of RE-induced EVs in other tissues, to better understand their contribution to whole-body adaptations to physical activity.

The reliability of EV studies is influenced by numerous factors, ranging from sample collection to the methods used for detection. Among these, one of the most critical steps is the isolation of EVs. Currently, the most widely adopted approaches involve combining different isolation techniques to maximize both yield and purity. Although a universally ideal method for EV isolation has yet to be established, a recent study by György et al. [142] introduced a promising protocol that combines SEC with UC. This combined method significantly reduced the presence of non-vesicular particles in the isolates, representing a potential step forward in the standardization of EV isolation procedures.

## 10. MiRNA Detection Methods

The accurate detection of miRNAs is influenced by numerous factors, ranging from sample collection protocols and isolation procedures to the detection methodologies employed. In the studies compiled within this review, substantial heterogeneity is observed among results, hindering direct comparisons across different investigations. Currently employed methods for miRNA detection include qPCR, in situ hybridization, microarray analysis, and RNA sequencing [143]. It is common that classical PCR-based methods are limited to the detection of only a few pre-selected miRNAs, not fully reflecting the complex molecular processes occurring in vivo. In contrast, high-throughput technologies such as NGS enable comprehensive profiling of the entire miRNA expression landscape. Table 1, Table 2 and Table 3 detail the miRNA detection methods employed, facilitating accurate and reliable interpretation of the obtained data. Compared to universal reverse transcription, the stem-loop primer-based approach offers greater specificity; however, it is typically restricted to reverse transcribing one miRNA per reaction [144]. To address this limitation, multiplexed stem-loop primer pools have been developed [145]. TaqMan miRNA assays utilize stem-loop RT primers, providing higher specificity and sensitivity during the reverse transcription of miRNAs [146]. In contrast, SYBR Green-based methods (e.g., miScript, mirVana) often employ less specific primers, increasing the likelihood of non-specific amplification or overlapping products. Therefore, the TaqMan technique is generally considered more reliable and accurate for miRNA detection. Microarray-based approaches encompass both multiplex qPCR arrays and hybridization-based arrays. In qPCR microarrays, pre-designed primer/probe sets are pre-loaded into 96- or 384-well plates, enabling parallel amplification and detection of multiple miRNAs. Hybridization-based arrays offer the advantage of enabling high-throughput, parallel analysis of numerous targets within a single sample at relatively low cost [143]. High-throughput small RNA sequencing enables quantitative identification of all small RNA species within a given sample, facilitating the discovery of novel miRNAs and other small non-coding RNAs [144].

## 11. Discussion

Changes of the miRNA profile caused by PA arouse the interest of many researchers, since these molecules play a significant role in the fine-tuning of protein synthesis, thus in the adaptation following exercise. Most of the publications collected here describe significant changes in the miRNA profile isolated from skeletal muscle, appearing in circulation or transported in EVs because of RE. However, based on the studies published so far, drawing general conclusions is extremely challenging, as both the studied population (young, elderly, healthy, ill, trained, untrained) and the study designs (exercise program, supplementary interventions, method of sample collection) exhibit extraordinary diversity. Future studies should focus specifically on miRNA profiles isolated from EVs if the aim is to investigate targeted intercellular communication, rather than assessing the total ci-miRNA profile. Nevertheless, it is evident that miR-1 and miR-133a showed varying expression in most studies (in plasma and muscle biopsies), as it was included among the selected few miRNAs in the majority of cases and may play a crucial role in the adaptation following RE. A significant part of differentially expressed miRNAs in response to RE are involved in the regulation of signaling pathways promoting hypertrophy or capillarization, so understanding these mechanisms following RE would have a huge practical impact on training protocols. The role of adipocyte lipolysis has been predominantly emphasized in the context of AER. However, in response to RE, EVs carrying miR-1 may contribute to the regulation of lipolysis [93,116], indicating that beyond the well-established hormonal regulation, miR-1 delivered via EVs may play an enhancing role in this metabolic process [93]. Since one miRNA can target many mRNAs and one mRNA can be regulated by several miRNAs, small changes in research protocols can greatly impact miRNA expression and function. Nutrient timing following RE has not yet been investigated in terms of miRNA and/or EV profile, which might be the goal of future research, as properly timed nutrient intake might be very important to maximize adaptation following RE. In the reviewed studies, differences in participant demographics, inclusion criteria, and outcome measures can introduce significant variability. The diversity in RE regimens, including intensity, duration, and frequency, further augments complexity. Due to the variability in exercise protocols, it is challenging to isolate the specific effects of RE on miRNAs and EVs, hindering the achievement of reliable and conclusive results. For EV studies in sport sciences, it would be advisable to follow the current MISEV guidelines [20], but these principles have also been summarized from the perspective of PA [147]. As the data (Table 1, Table 2 and Table 3) show, the miRNA responses following RE, apart from individual diversity, are very diverse and often contradictory due to different testing approaches, sampling, isolation and detection methods. One might assume that as a result of RE, the muscle as an endocrine organ would be involved in the production of EVs; however, a significant portion of the publications considered led to the conclusion that muscle adaptation takes place locally and not through EVs entering circulation, although miRNAs derived from EVs may have a positive effect in other tissues. It is exciting to consider EV-transported miRNAs for curing muscle diseases, enhancing muscle adaptation, or influencing lipid metabolism, though this will only be possible after fully understanding the molecular cascades influenced by EV cargoes. Nevertheless, after RE, EVs can play an important role in adaptation and disease prevention, requiring further studies. The publications collected here can provide a basis for further research to interpret results. By delving into the role of EVs and miRNAs in muscle adaptation, trainers gain valuable insights enabling them to tailor exercises in accordance with the specific requirements of each individual, objectives, and any potential physical limitations they may have. The insights gleaned from the articles hold significant value for clinicians who work closely with populations vulnerable to muscle deterioration, such as older adults or individuals with specific medical conditions. By incorporating strategies grounded in RE and ensuring optimal nutrient intake, clinicians can effectively mitigate or decelerate muscle loss within these groups. Fitness professionals and clinicians can use biomarkers such as EVs and miRNAs as indicators of progress and adaptation to RE. The review may also inspire further research into the role of EVs and miRNAs in exercise physiology. Future studies could explore additional biomarkers, mechanisms, or interventions to optimize the benefits of RE for various populations and health conditions.

## 12. Conclusions

Current findings highlight the complex and highly variable nature of miRNA and EV responses to RE, influenced by differences in study populations, protocols, and methodologies. Despite these inconsistencies, miRNAs, particularly miR-1 and miR-133a, emerge as potential key regulators in muscle adaptation and metabolic processes following RE. Understanding the specific role of EV-derived miRNAs in intercellular communication could open new avenues for personalized training strategies, clinical applications, and therapeutic interventions. Future research should aim for standardized protocols and focus on targeted EV analysis to clarify the molecular mechanisms underlying adaptation to RE.

## Figures and Tables

**Figure 1 cimb-47-00583-f001:**
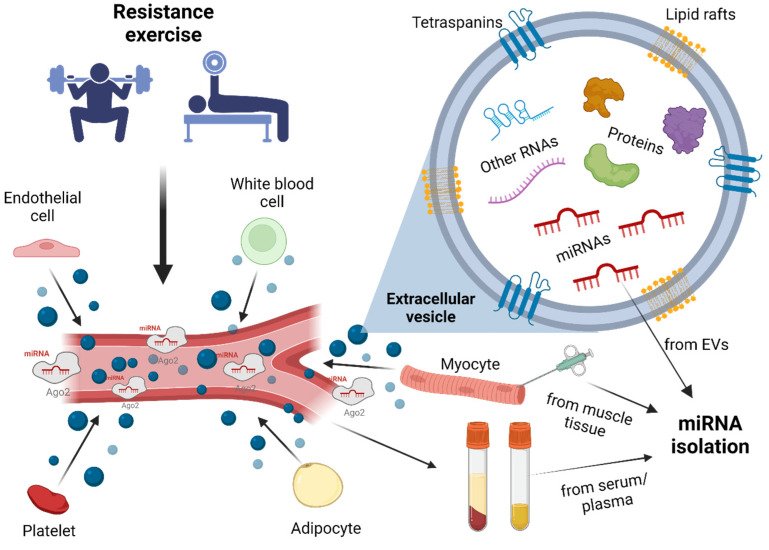
The effect of RE on the circulating miRNA profile. miRNAs found in circulation are bound to proteins: ago2 (the silencing complex and the mature miRNA) or packaged in extracellular vesicles. Different cell types (endothelial cells, myocytes, white blood cells, platelets, fat cells) can express different types of miRNAs (bound to proteins, packaged in EVs), which might influence the functioning of other cells, promoting cell–cell communication. Various proteins (tetraspanins: CD9, CD63, CD81) or lipid rafts (cholesterol, flotillin) are embedded into the membrane of EVs, while miRNAs, other nucleic acids and proteins (Alix, TSG101, Chaperones, signaling proteins) can be transported in their lumen. Isolation of miRNA can be done from blood (plasma, serum), or muscle tissue (biopsy) samples, whereas EVs are isolated from plasma. Created with BioRender.com.

**Table 1 cimb-47-00583-t001:** The effect of acute RE on the miRNA profile. MiRNA changes are only within-group compared to baseline. RE: resistance exercise; RM: repetition maximum; BFR: Blood flow restriction; yr: year; RT-PCR: Real-time polymerase chain reaction; T2DM: Type 2 diabetes mellitus; HI: high intensity; LI: low intensity.

Reference	Subjects	Exercise Stimulus	Additional Intervention	Samples	MiRNA Detection Method	MiRNA Responses
Drummond et al., 2008 [78]	Healthy, untrained young (n = 6, age 29 ± 2 yr) and elderly men (n = 6, age 70 ± 2 yr)	Acute RE: leg extension 10 reps of 8 sets 70% at 1RM	20 g of EAA ingestion 1 h after the training session	Muscle biopsies (*Vastus lateralis*): pre, 1 h, 3 h, 6 h	Pri-miRNA: SYBR Green RT-PCR miRNA: mirVana SYBR Green RT-PCR	Young: 3 h: 1↓ 6 h: Pri-206↑ Pri-1-2, 1, Pri-133a-1, Pri-133a-2↓ Elderly: 3 h: Pri-206↑ Pri-133a-2↓
Sawada et al., 2013 [72]	Healthy, recreationally active men (n = 12), age 29 ± 1.2 yr	Acute RE: Bench press and bilateral leg press, 5 sets of 10 reps at 70% 1RM		Blood (serum), pre, immed, 1 h, 1 day, and 3 days after exercise	miRCURY™ LNA miRNA Array, TaqMan MicroRNA assay RT-PCR	3 days after: 149↑ 146a, 221↓
Rivas et al., 2014 [79]	Healthy young (n = 8, age 22 ± 1 yr) and elderly men (n = 8, age 74 ± 2 yr)	Acute RE: bilateral knee extension, bilateral leg press of 3 sets 80% at 1RM	Young and elderly men comparison	Muscle biopsies (*Vastus lateralis*): pre, 6 h	miScript SYBR Green PCR Array	Young: 423-5p↑ 16-5p, 23b-3p, 24-3p, 26a-5p, 26b-5p, 27a-3p, 27b-3p, 29a-3p, 29c-3p, 30a-5p, 30d-5p, 95-3p,107, 126-3p, 133a, 133b, 140-3p, 181a-5p, 324-3p, 378a-5p↓ Elderly: 423-5p↑
Uhlemann et al., 2014 [71]	Healthy, trained subjects (n = 11)	Acute RE: Lat pulldown, leg press and butterfly, 3 sets of 15 reps, additional eccentric load		Blood (plasma) pre, immed, 1 h	TaqMan miRNA Assay RT-PCR	Immed. after RE: 133↑
Zacharewicz et al., 2014 [80]	Healthy, untrained young (n = 10, age 24.2 ± 0.9 yr) and elderly men (n = 10, age 66.6 ± 1.1 yr)	Acute RE: leg extension, 14 reps of 3 sets 60% at 1RM	Young and elderly men comparison	Muscle biopsies (*Vastus lateralis*): pre, 2 h	TaqMan Array Human MicroRNA A + B Cards v3.0	26 miRNAs were regulated with age and/or exercise, 7 of these were differentially and the other 7 were regulated in either young or old subjects. Young: 486-3p↑ 149-3p, 99b-5p, 520g-5p↓ Old: 99a-5p↑ 196b-5p, 489-5p, 628-5p, 186-5p, 335-5p↓
Ogasawara et al., 2016 [81]	Healthy, untrained young men (n = 10)	Acute RE: bilateral knee extension and flexion, 10 reps of 3 sets at 70% of 1RM	High and low responders comparison	Muscle biopsies (*Vastus lateralis*): pre, 3 h after	NanoString nCounter human miRNA expression assay	84 miRNAs differentially expressed
Cui et al., 2017 [67]	Healthy, active, young men with no gym experience (n = 15), age 19.36 ± 0.14 yr	Acute RE: Strength endurance: bench press, squat, pulldown, overhead press, standing dumbbell curl, 3 sets of 16–20 reps at 40% 1RM		Blood (plasma), pre, immed, 1 h, 24 h	TaqMan Low-Density Array, TaqMan RT-PCR	1 h after: 532↑ Immed after exercise: 208b↓
Cui et al., 2017 [67]	Healthy, active, young men with no gym experience (n = 15), age 19.72 ± 0.20 yr	Acute RE: Muscular hypertrophy: bench press, squat, pulldown, overhead press, standing dumbbell curl, 3 sets of 12 reps at 70% 1RM		Blood (plasma), pre, immed, 1 h, 24 h	TaqMan Low-Density Array, TaqMan RT-PCR	Immed after exercise: 21, 133a↓ 1 h after: 181a, 206↑ 221↓ 24 h after: 133b↑
Cui et al., 2017 [67]	Healthy, active, young men with no gym experience (n = 15), age 18.87 ± 0.12 yr	Acute RE: Maximum strength: bench press, squat, pulldown, overhead press, standing dumbbell curl, 4 sets of 6 reps at 90% 1RM		Blood (plasma): pre, immed, 1 h, 24 h	TaqMan Low-Density Array, TaqMan RT-PCR	Immed after exercise: 133a↓ 1 h after: 133b↑
D’Souza et al., 2017 [76]	Healthy, resistance-trained, young men (n = 9), age 24.6 ± 4.9 yr	Acute RE: Leg press, 2 sets of 10 reps 50–70% at 1 RM, 6 sets of 8–10 reps 80% at 1 RM, leg extension, 8 sets of 8–10 reps 80% at 1RM		Blood (plasma), muscle biopsies (*Vastus lateralis*): pre, 2 h, 4 h	TaqMan Advanced miRNA RT-PCR	Muscle: 2 h after: 133a, 206↑ 23a, 378b↓ 4 h after: 486, 146a↑ 23a↓ Blood: 4 h after: 133a, 149↑
Margolis et al., 2016 [73]	Healthy, inactive, young (n = 9, age 22 ± 1) and old (n = 9, age 74 ± 2 yr)	Acute RE: Bilateral leg extension and leg press, 3 sets of 10 reps at 80% 1RM		Blood (serum), pre, immed, and 6 h after	miScript miRNA PCR Array SYBR Green, TaqMan MicroRNA Assays	6 h after exercise in young: 17-5p, 19a-3p, 19b-3p, 20a-5p, 26b-5p, 93-5p, 106-5p, 143-3p, 195-5p↑
Morais Junior et al., 2017 [82]	T2DM and healthy age 68.2 ± 5.3 yr men and women (n = 23)	Acute strength exercise: circuit fashion (8 exercises) 3 sets 40 s and 20 s	Strength and cardiovascular training circuit comparisons	Blood (serum): pre, post	TaqMan RT-PCR	146a-5p↑
Russell et al., 2017 [83]	Healthy young (n = 10, age 18–30) and elderly (n = 10, 60–75) men	Acute RE: leg extension, 3 sets of 14 reps	Young/elderly comparisons	Muscle biopsies (vastus lateralis): pre, 2 h	TaqMan Array Human MicroRNA A + B Cards version Set v3.0	26 miRNAs that were significant (previously reported Zacharewicz et al., 2014 [80]) young: 520g-3p, 628-5p↓
Bjørnsen et al., 2019 [84]	Recreationally active age 24 ± 2 yr men and women (n = 13)	Acute RE: two 5-day blocks of 7 BFRRE sessions, separated by a 10-day rest period. 4 sets of unilateral knee extensions to voluntary failure at 20% of 1RM	Partial BFR	Muscle biopsies (vastus lateralis): pre, during: Acute 1, Day 4, Rest week, Acute 2, 3 days, and 10 days after	TaqMan Advanced miRNA RT-PCR	Rest week: 208b↓ Acute2: 208b, 486↓ Post 10: 16, 486↑
D’Souza et al., 2019 [85]	Healthy, recreationally active elderly men (n = 23) age 67.9 ± 0.9 yr	Acute RE: bilateral barbell smith racks squat, 45° leg press, seated knee extensions, 8–10 reps of 3 sets 80% at 1RM (circuit manner)	Protein ingestion: placebo, 20 or 40 g whey protein	Muscle biopsies (*Vastus lateralis*): pre, 2 h and 4 h	TaqMan Advanced miRNA RT-PCR	16-5p was altered in all groups placebo: 4 h: 15a, 499a↑ 40g: 2 h, 4 h: 451a↑
Vogel et al., 2019 [75]	Healthy men and females (n = 18) age 25 ± 2 yr	Acute low intensity (LI) RE: leg flexion/extension, total 75 reps in 4 sets 30% at 1RM	BFR	Blood (plasma): pre, immed	RT-PCR: miRCURY LNA miRNA PCR assays	143-3p↓
Vogel et al., 2019 [75]	Healthy men and females (n = 18) age 25 ± 2 yr	Acute RE: leg flexion/extension, total 75 reps in 4 sets 30% at 1RM		Blood (plasma): pre, immed	RT-PCR: miRCURY LNA miRNA PCR assays	
Vogel et al., 2019 [75]	Healthy men and females (n = 18) age 25 ± 2 yr	Acute RE: leg flexion/extension, total 30 reps in 3 sets 70% at 1RM		Blood (plasma): pre, immed	RT-PCR: miRCURY LNA miRNA PCR assays	10b-5p, 30a-5p, 139-5p, 143-3p, 195-5p↑
Hashida et al., 2021 [77]	Physically inactive men (n = 7)	LI RE		Blood (serum): pre, post	miRNA microarray	7 miRNAs significantly changed, of these: 630, 5703↑
Telles et al., 2021 [86]	Healthy, untrained young men (n = 9) age 23.9 ± 2.8 yr	Acute RE: leg press, leg extension, 8–12 reps of 2 sets until muscle failure, high intensity (HI) interval exercise: 12 × 1-min sprints, RE and HI interval exercise combined		Muscle biopsies (*Vastus lateralis*): pre, immed, 4 h, 8 h	TaqMan Advanced miRNA RT-PCR	RE: 1-3p, 23a-3p, 133-a-3p, 133-b. 181a-3p, 206, 486↑
Torma et al., 2021 [87]	Healthy men (n = 7) age 24.5 ± 4.7 yr	Acute RE: squats, 10 reps of 7 sets 70% at 1RM	BFR during rest periods	Muscle biopsies (*Vastus lateralis*): 2 h	TaqMan miRNA RT-PCR	BFR leg: 206↓
Buchanan et al., 2022 [88]	Postmenopausal women (n = 10) age 65–76 yr	Acute RE: 3 sets of 10 repetitions per exercise at 70–75% of 1RM: leg press, shoulder press, lat pulldown, leg extension, and hip adduction	RE and whole body vibration comparison	Blood (serum): pre, post, 1 h, 24 h, 48 h	SYBR Green RT-PCR	RE: no changes
D’Souza et al., 2023 [89]	Physically active young 22 ± 2 yr men (n = 9)	Acute RE: unilateral knee extension, unilateral 45° leg press, 6 sets (8, 8, 10, 12, 10, and 10 reps), single-leg squats and walking lunges, 3 sets (12 reps)	Cold water immersion, or active recovery after RE	Muscle biopsies (*Vastus lateralis*): pre, 2 h, 24 h, 48 h	TaqMan Advanced miRNA RT-PCR	Cold water immersion: 24 h: 133a, 126↑ 48 h: 126↑ active recovery: 48 h: 1↑
Benavente et al., 2024 [90]	Strength-trained males aged (Normoxia 22.7 ± 3.4 yr, Hipobaric hypoxia 22.8 ± 4.2 yr, Normobaric hypoxia 21.9 ± 2.2 yr) (n = 33)	Acute and Chronic RE: full body routine, 6 exercises, 3 sets of 6–12 reps at 65–80% of 1RM	Normoxia, Hypobaric hypoxia, Normobaric hypoxia	Blood (serum): 72 h pre, Post first RE, Post last RE	SYBR Green RT-PCR	Post first RE: Hypobaric hypoxia: 206↓ Normobaric hypoxia: 206↑ Post last RE: Normoxia: 206↑ Hypobaric hypoxia: 206↑ Normobaric hypoxia: 206↑
Takamura et al., 2024 [91]	Healthy, untrained males, age LI 26.17 ± 4.40 yr, HI 25 ± 3.2 yr (n = 12)	Acute RE: leg extension and leg curl, 5 sets of 10 reps at 10% of 1RM (LI), or 80% of 1RM (HI)	LI and HI exercise	Blood (plasma): pre, post	TaqMan RT-PCR	HI: 195↑ LI: 29c, 486↑

**Table 2 cimb-47-00583-t002:** Publications so far investigating the effect of chronic RE on the miRNA profile. MiRNA changes are only within-group compared to baseline. RE: resistance exercise; RM: repetition maximum; BFR: Blood flow restriction; yr: year; RT-PCR: Real-time polymerase chain reaction; NGS: Next-generation sequencing; AER: aerobic exercise; T2DM: Type 2 diabetes mellitus; MH: muscle hypertrophy; HI: high intensity.

Reference	Subjects	Exercise Stimulus	Additional Intervention	Samples	MiRNA Detection Method	MiRNA Responses
Davidsen et al., 2011 [100]	Healthy young, physically active men (n = 56), age 18–30 yr	12-week training, 5 sessions/week, pushing, pulling, and leg exercises	High and low responders comparison	Muscle biopsies (*Vastus lateralis*): 48 h pre and post after first/last training session	TaqMan RT-PCR	Low responders: 26a, 29a, 378↓ 451↑
Mueller et al., 2011 [101]	Elderly (age 80.1 ± 3.7 yr) men and women RE (n = 13), eccentric ergometer (n = 14)	12-week, 2 sessions/week, leg press, knee extension, leg curl, hip extension, 3 sets of 8–10 reps	RE or eccentric ergometer sessions	Muscle biopsies (*Vastus lateralis*): pre, post	miScript primer assay SYBR Green	Both groups: 1↓
Rowlands et al., 2014 [102]	Inactive, Polynesian T2DM males and females age 49 ± 5 yr (n = 17)	16-week training, 3 sessions/week: two or three sets of eight exercises using machine weights, six to eight repetitions to fatigue	RE and endurance exercise comparison	Muscle biopsies (*Vastus lateralis*): pre, post	Affymetrix GeneChip microarray, TaqMan miRNA assay RT-PCR	23a, 195, 3178, 483-5p, 487↑ 193b, 1207-5p↓
Zhang et al., 2015 [103]	Sedentary older men (n = 3) and women (n = 4), age 65–80 yr	5 months, 3 sessions/week: leg press, knee extension, leg curl, calf press, 70% at 1RM		Muscle biopsies (*Vastus lateralis*): pre, 5 months after; Blood (plasma): pre, 5 months after	TaqMan miRNA assay RT-PCR	Muscle: 133b↓
Ogasawara et al., 2016 [81]	Healthy, untrained young men (n = 18), age 21.4 ± 1.1 yr	12-week training, 3 sessions/week: knee extension, flexion, 10 reps of 3 sets, 70% at 1RM	High and low responders comparison	Muscle biopsies (*Vastus lateralis*): pre, 12 w after	NanoString nCounter human miRNA expression assay	102 miRNAs differentially expressed
D’Souza et al., 2018 [98]	Physically active young, resistance exercise-trained men, age 21.5 ± 0.6 yr (n = 21)	12-week training, 2 sessions/week: Bilateral 45° leg press, knee extension, walking, lunges, plyometrics exercises	Cold water immersion, or active recovery after RE; Whey protein isolate consumed 1 h before and following the completion of therapy, plus recovery bar with 18 g of protein and 30.7 g of carbohydrate was also consumed 2 h post-training	Muscle biopsies (*Vastus lateralis*): pre, post	TaqMan Advanced miRNA RT-PCR	Active recovery: 15a, 16, 208b, 499a↑
Hagstrom and Denham 2018 [104]	Breast cancer survivors (women, n = 24)	16-week, 3 sessions/week	RE or usual care intervention	Blood (serum): pre, 16 w after	RT-PCR	No change, only between high and low responders: miR-133a-3p, miR-370-3p↑
Horak et al., 2018 [105]	Healthy young men (n = 30), age 22.5 ± 4.06 yr	Explosive strength training, MH and HI interval exercise		Blood (plasma): pre, 5 week after, post	TaqMan miRNA assay RT-PCR	Explosive strength training: 222, 16↓ MH: 93, 16, 222↓
Gazova et al., 2019 [106]	Sedentary prostate cancer patients (men, age 69.21 ± 5.8 yr, n = 15)	Strength training: 16-week, 3 times/week	Exercise and control group comparisons	Blood (plasma): pre, post	TaqMan miRNA assay RT-PCR	1, 29, 133↑
Olioso et al., 2019 [107]	Elderly, sedentary individuals with type 2 diabetes mellitus (n = 6: AER = 3; RE = 3), age 40–70 yr	AER or RE: 60 min, 3 times per week, for a period of 4 months, RE: lower, upper body, core exercises, 70–80% at 1RM		Blood (plasma): pre, after exercise training	c-miRNA PCR panel (Exiqon), TaqMan RT-PCR	Irrespective of AER/RE: 423-3p, 451a, 766-3p↑
Schwarz et al., 2019 [108]	Healthy, recreationally active resistance-trained young men (n = 16), age 22.5 ± 3.1 yr	4-week periodized training, consisting of 2 lower-body and 2 upper-body sessions. 7 exercises/session.	Pre-Workout drink ingestion: 26.1 g Bang Master Blaster; 26.1 g placebo	Muscle biopsies (*vastus lateralis*): pre, post 4 weeks	SYBR Green RT-PCR	Both groups: 23a, 23b↑
Liu et al., 2020 [109]	Healthy older men and females (n = 10), age 67.6 ± 2.2 yr	12-week training		Blood: pre, after 12 weeks	Illumina NextSeq NGS	Adipogenesis-related: 103a-3p, 103b, 143-5p, 146b-3p, 146b-5p, -17-5p, 181a-2-3p, 181b-5p, 199a-5p, 204-3p, and -378c anti-adipogenesis-related: 155-3p, 448, 363-3p myogenesis-related: 125b-1-3p, 128-3p, 133a-3p, 155-3p, 181a-2-3p, 181b-5p, 199a-5p, 223-3p, 499a-5p inflammation-related: 146b-3p, 146b-5p, 155-3p, 181a-2-3p, 181b-5p
Banitalebi et al., 2021 [110]	Untrained females with Osteosarcopenic Obesity (n = 63), age 65–80 yr	12-week training, 3 sessions/week using elastic bands		Blood (serum): pre, after 12 weeks	SYBR Green RT-PCR	133, 206 not changed
Estébanez et al., 2021 [111]	Healthy elderly men and females (n = 38), age 70–85 yr	8-week training, 2 sessions/week, leg press, ankle extension, bench press, leg extension, biceps curl, pec deck, high pulley traction, dumbbell lateral lift, 3 sets of 12-8-12 repetitions		Blood (plasma): pre, after 12 weeks	TaqMan RT-PCR	146a-5p not changed
Rivas et al., 2021 [99]	Inactive, elderly males and females, age 70–85 yr (n = 73), Losers (78 ± 6), Gainers (77 ± 6)	6 months training, 3 sessions/week: leg press, seated row, leg extension, chest press, and leg curl at 80% of 1RM, 2 sets of 10 then 3 sets of 12 reps	Whey protein supplement or isocaloric control beverage consumption, twice a day. Leg lean mass losers (Losers) or gainers (Gainers) comparison	Blood (serum): pre, post	TaqMan RT PCR, miRCURY LNA SYBR, Green RT-PCR	No pre, post comparison, only between groups: 19b-3p, 92a, 126, 133a-3p, -133b↑, -1-3p↓ was in Gainers, compared to Losers
Torma et al., 2021 [112]	Healthy young men (n = 22), age control: 23.9 ± 1.7, BFR: 24.1 ± 6.1	4-week training, 3 sessions/week, 10 reps of 5 sets, 70% at 1RM	BFR during rest periods	Muscle biopsies (*vastus lateralis*): 72 h pre, 24 h post after first/last training session	TaqMan RT-PCR	BFR: 1, 133a↓
Corrêa et al., 2022 [113]	Older hemodialysis patients (n = 25), age 68 ± 1, men and women	24-week, 3 sessions/week	RE (n = 13) or control (n = 12)	Blood	TaqMan RT PCR	RE: 1↑, 31↓
Agostini et al., 2023 [114]	Elderly (age > 60 yr) frail (n = 15) and robust (n = 30) males and females	Multicomponent exercise: 12-week, RE, gait training, balance training	Frail and robust subjects, comparisons	Blood (serum): pre, post	Digital Droplet PCR	Both groups: 93-5p, 495-3p↓, 155-5p↑

**Table 3 cimb-47-00583-t003:** Publications examining the effect of RE on miRNAs delivered in EVs. miRNA changes only appear as within-group changes compared to baseline. These changes are greatly influenced by the isolation and detection method used, in addition to other differences in testing approaches. DUC: Differential ultracentrifugation; UC: Ultracentrifugation; SEC: Size exclusion chromatography; RE: resistance exercise; IB: Immunoblotting; NTA: Nanoparticle tracking analysis; TEM: Transmission electron microscopy; WB: Western blotting; EV: extracellular vesicles; RT-PCR: Real-time polymerase chain reaction; NGS: Next-generation sequencing.

Reference	Subjects	Exercise Stimulus	Samples	EV Isolation Method	EV Detection Method	miRNA Detection Method	MiRNA Responses
Lovett et al., 2018 [122]	Healthy, untrained men (n = 9), age 18–30 yr	Acute plyometric jumps, 10 sets, 10 reps, followed by 5 sets of 4 min downhill running	Blood (plasma): pre, 2 h, 24 h	SEC	TEM, NTA	TaqMan Advanced RT-PCR	24 h after: 31↓
Annibalini et al., 2019 [65]	Healthy, resistance exercise-trained young men (n = 8), age 23.7 ± 2.8 yr	Acute flywheel RE: squats, 5 sets, 10 maximal reps	Blood (plasma): pre, 2 h, 24 h, 48 h, Muscle biopsies: pre, 2 h	DUC	NTA	SYBR Green RT PCR	2 h after: 206, 146a↑
Just et al., 2020 [24]	Healthy, recreationally active young men (n = 9), age 21 ± 0.6 yr	Acute blood flow restricted RE: knee extensions, 5 sets, volitional failure 30% at 1RM	Blood (plasma): pre, 1 h	precipitation, SEC	NTA, WB, TEM, EV Array	Illumina NextSeq NGS	1 h after: 7b-5p, 16-5p, 182-5p, 363-3p, 451a-5p, 1294↑ 17-5p, 19b-3p, 21-5p, 150-5p, 221-3p, 340-5p↓
Vechetti et al., 2021 [116]	Healthy, recreationally active males and females (n = 10), age 26–50	Acute RE: leg press, knee extension, 4 sets, 7 reps	Blood (plasma): pre, 30 min	magnetic beads	NTA, TEM, WB, EV labeling and tracking	TaqMan RT PCR	30 min after: 1↑
Xhuti et al., 2023 [123]	Older adults (n = 19), age 74.9 ± 5.7 yr	Chronic home-based RE (12 weeks): 3 sets, 10–15 reps	Blood (plasma): pre, post 12 weeks, Muscle biopsies: pre, post	SEC-UC	IB, NTA, TEM	TaqMan RT PCR	23a, 27a, 146a, 92a↑
Kawanishi et al., 2023 [124]	C57BL/6 male mice (n = 18), age 10 wk	Acute EPS-induced RE: 6 sets, 10 contractions for 3 s	Blood (serum): pre, post, 1.5 h	precipitation	NTA, WB, EV Quantification Assay	Ion Torrent NGS, TaqMan RT PCR	1,5 h after: 1a-3p, 133a-3p, 206-3p↑
Conkright et al., 2024 [125]	Participants (n = 10), age 26.9 ± 5.5 yr	Acute RE: back squat, 6 sets, 10 reps, 75% at 1RM	Blood (plasma): pre, post	SEC		Illumina NextSeq NGS	34 miRNAs were altered
Burke et al., 2024 [93]	Healthy, untrained males and females (n = 32), age 29.2 ± 6.2 yr	Acute RE: back squat, leg press, knee extension, and latissimus pulldown, 3 sets, 8 reps at 80% 1RM, 4th set continued until failure	Blood (serum): pre, post, 30 min, 1 h, 90 min Muscle biopsies (vastus lateralis): pre, 45 min Adipose tissue biopsies: 70 min	SEC	ExoView	TaqMan RT PCR	Serum: post, 90 min: 1↑ Muscle: pri-miR-1a↑ Adipose: 1↑

## Data Availability

The electronic bibliographic database PubMed was used to search for references.

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
