# Peer review of "The Role of miRNAs and Extracellular Vesicles in Adaptation After Resistance Exercise: A Review"

_cimb, 2025, doi:10.3390/cimb47080583_

Round 1

Reviewer 1 Report

Comments and Suggestions for Authors

Márta Wilhelm and co-authors submitted a review to CIMB devoted to the role of muscular miRNAs and extracellular vesicles in exercise

The reviewer has several recommendations that should be resolved before the article can be recommended for publication

In Tables 1 and 2, RT-PCR is indicated as the method for detecting miRNAs, there are many approaches for detecting miRNAs, for example, stem-loop PCR, some of which are more reliable, some are not. It is necessary to clarify in the tables what form of qPCR the authors used. The same applies to NGS: i.e. which platform and method of library preparation were used.

A section could also be added to describe the approaches for detecting miRNAs and their strengths and weaknesses.

The Conclusions section is too long for a review article. I think it should be shortened to one short paragraph of 2-3 sentences. In its current form, it corresponds more to the Discussion section

Sincerely,

Author Response

Summary

Thank you very much for taking the time to review this manuscript. Please find the detailed responses below and the corresponding corrections highlighted in the re-submitted files.

Comments 1: “In Tables 1 and 2, RT-PCR is indicated as the method for detecting miRNAs, there are many approaches for detecting miRNAs, for example, stem-loop PCR, some of which are more reliable, some are not. It is necessary to clarify in the tables what form of qPCR the authors used. The same applies to NGS: i.e. which platform and method of library preparation were used.”

Response 1: Thank you for pointing this out; we fully agree with your comment! Therefore, we have added a new section emphasizing the importance of stem-loop PCR, which provides valuable context for interpreting the tables. Additionally, we clarified the different detection methods within the tables. In Table 3, we included a new column specifically detailing the miRNA detection techniques.

Comments 2: “A section could also be added to describe the approaches for detecting miRNAs and their strengths and weaknesses.”

Response 2: Thank you for the valuable suggestion, we agree that including such a section enhances the scientific quality of the manuscript and provides useful context for other researchers in interpreting the data. Accordingly, we have added a new subsection titled "MiRNA detection methods", highlighted in yellow in the revised manuscript.

10. MiRNA detection methods

The accurate detection of miRNAs is influenced by numerous factors, ranging from sample collection protocols and isolation procedures to the detection methodologies employed. In the studies compiled within this review, substantial heterogeneity is observed among the results, which complicates direct comparisons across different investigations. Currently employed methods for miRNA detection include qPCR, in situ hybridization, microarray analysis, and RNA sequencing [143]. It is common that classical PCR-based methods are limited to detecting only a few pre-selected miRNAs, which may not fully reflect the complex molecular processes occurring in vivo. In contrast, high-throughput technologies such as NGS enable comprehensive profiling of the entire miRNA expression landscape. Tables 1, 2, and 3 detail the miRNA detection methods employed, facilitating accurate and reliable interpretation of the obtained data. Compared to universal reverse transcription, the stem-loop primer-based approach offers greater specificity; however, it is typically restricted to reverse transcribing one miRNA per reaction [144]. To address this limitation, multiplexed stem-loop primer pools have been developed [145]. TaqMan miRNA assays utilize stem-loop RT primers, which provide higher specificity and sensitivity during the reverse transcription of miRNAs. In contrast, SYBR Green-based methods (e.g., miScript, mirVana) often employ less specific primers, increasing the likelihood of non-specific amplification or overlapping products. Therefore, the TaqMan technique is generally considered more reliable and accurate for miRNA detection . Microarray-based approaches encompass both multiplex qPCR arrays and hybridization-based arrays. In qPCR micro-arrays, pre-designed primer/probe sets are pre-loaded into 96- or 384-well plates, enabling parallel amplification and detection of multiple miRNAs. Hybridization-based arrays offer the advantage of enabling high-throughput, parallel analysis of numerous targets within a single sample at relatively low cost [143]. High-throughput small RNA sequencing enables quantitative identification of all small RNA species within a given sample and facilitates the discovery of novel miRNAs and other small non-coding RNAs [144].

Comments 3: “The Conclusions section is too long for a review article. I think it should be shortened to one short paragraph of 2-3 sentences. In its current form, it corresponds more to the Discussion section.”

Response 3: Thank you for the suggestion! We fully agree that the conclusion should be shorter and more concise, and that the previous conclusion section was more appropriate as part of the discussion. Therefore, we have renamed the former “Conclusion” section to “Discussion,” and inserted a new, brief “Conclusion” section afterward, highlighting the key findings and main message of the manuscript.

11. Conclusion

Current findings highlight the complex and highly variable nature of miRNA and EV responses to RE, influenced by differences in study populations, protocols, and methodologies. Despite these inconsistencies, miRNAs, particularly miR-1 and miR-133a, emerge as potential key regulators in muscle adaptation and metabolic processes following RE. Understanding the specific role of EV-derived miRNAs in inter-cellular communication could open new avenues for personalized training strategies, clinical applications, and therapeutic interventions. Future research should aim for standardized protocols and focus on targeted EV analysis to clarify the molecular mechanisms underlying adaptation to RE.

We would like to express our sincere gratitude for your thorough and detailed review of our manuscript! Your insightful comments and suggestions, both in terms of content and formatting, have significantly contributed to improving the clarity, structure, and overall scientific value of the paper. We believe that the revisions made in response to your feedback have substantially strengthened the manuscript, and we are truly grateful for the time and effort you dedicated to its careful evaluation!

Reviewer 2 Report

Comments and Suggestions for Authors

Review on the manuscript titled “The role of miRNAs and extracellular vesicles in adaptation after resistance exercise: a review” by Csala et al., 2025.

                The review assesses the impact of miRNA in adaptation after training exercise. In their abstract the authors mention “Extracellular vesicles and encapsulated miRNAs are fine tuners of protein synthesis and are important in the adaptation after resistance training.”

                In the comprehensive introduction the authors stress the undoubted benefits of fitness exercise. In particular, the authors state that “Recently it was discovered that biomarkers like cytokines and extracellular vesicles (EVs), especially small EVs, are released into the circulation and serve in cell to cell and inter-tissue communication “. The authors’ aim was to elaborate on the EVs content in their review, stressing on miRNA molecules.

                In M&Ms the authors convey their strategy on studies compilation, that comprise 130 studies overall.

                The review contains several further chapters:

  1. Extracellular Vesicles. Here the authors elaborate on EVs after RE, resulting in fig. 1
  2. miRNAs. Herein, the authors describe miRNA biogenesis and function, as well as listing 8 striatal muscle-specific miRNAs: miR-1, -133a, -133b, -206, -208a, -208b, -486, -499.

4.2. Circulating microRNAs. In the bloodstream, ci-miRNAs are transported to the target cells with the aid of EVs (microvesicles, exosomes), proteins (Argonaute), or high-density lipoproteins (HDL), and might affect the translation of complementary mRNA.

                This chapter contains major Table 1 with explicit analysis of miRNA turnover after the acute ER. (25 studies).

  1. Ci-miRNAs after acute RE.

                Herein, the authors elaborate on studies listed in Table 1.  The authors note that “Most published researches so far have examined the miRNA profile changes after acute, aerobic exercise.”

  1. The relationship between acute RE and the microRNA profile from muscle biopsy

                Herein, the authors discuss the muscle biopsies studies listed in Table 1 (? Please reference the source). The authors note that “Muscle biopsies of powerlifters and untrained controls were compared at rest, find-ing 12 miRNAs differentially expressed [79]”. It’s not clear why [79] not included in any table.

  1. The miRNA profile after RE with protein consumption (Table 1? Are corresponding studies presented there? Please, reference the source). The authors note that this regime invokes Akt-mTOR channel with the consequent further responses.
  2. miRNA profile in response to Chronic RE studies: It’s implied that chronic RE have focused on the elderly population. The authors note that “Due to highly diverse research approaches so far, synthesizing the results and drawing general conclusions has been extremely challenging”. The data on Chronic RE studies is presented in Table 2 (18 studies).
  3. The effect of RE on extracellular vesicle (EV) profile changes. The data is presented in Table 3 (8 studies), and this is the longest chapter of the manuscript.
  4. Conclusions. The authors presented extensive discussion on the subject, stressing difficulties due to the heterogeneity of the studies, and various research designs.

                Overall, the manuscript may be of interest to the researchers/clinicians/sport managers in the field. Still, some flaws are listed in notes below.

  • 1: ago2 network comprises a lot more proteins, e.g.: AGO1, AGO2, DDX20, DDX6, DHX9, DICER1, FMR1, GEMIN4, TARBP2, TNRC6A, TNRC6B. Please, ascertain if there are all of them in the EV.
  • The authors state that “Different cell types (endothelial cells, myocytes, white blood cells, platelets, fat cells) can express different types of miRNAs (bound to proteins, packaged in EVs), which might influence the functioning of other cells, promoting cell-cell communication”. While the EV content may enter various cell types, it’s not clear if there are some specificity of cells target.
  • Chapter “4. miRNAs.” could be renamed as “muscle specific miRNA’
  • 76 : MiRNAs appearing in the blood circulation
  • Table 1 should be moved to Chapter 5.
  • ‘Chronic RE’ term should be discussed (elaborated) on the first instance.
  • The authors mention stress hormones involved in the physical training studies, which I think are the major players in the process. Notably, blood brain barrier doesn’t allow EVs to brain. Could the authors elaborate on this issue?
  • Conclusion section looks more like a discussion. It’s recommended to rename it, and provide a short conclusion section with major points.
  • Please, check if all of abbreviations in the table are present/absent in the text; some are not comprehensive: IB Im:

EXPL Explosive strength training

HI High-intensity

HIIE High-intensity interval exercise

HR High responders

HYP Hypertrophic strength training

IB Im (?)

LI Low-intensity

11. It's also unclear if all types of exersises mentioned are related to ER type, please, explain.

Author Response

Summary

Thank you very much for taking the time to review this manuscript! Please find the detailed responses below and the corresponding corrections highlighted in the re-submitted files.

Comments 1: “Herein, the authors discuss the muscle biopsies studies listed in Table 1 (? Please reference the source).
The authors note that “Muscle biopsies of powerlifters and untrained controls were compared at rest, finding 12 miRNAs differentially expressed [79]”. It’s not clear why [79] not included in any table.”

Response 1: Thank you for the comment! We have revised the paragraph to include a reference to Table 1.
This specific data was not included in the table because the tables present within-group changes in the miRNA profile before and after the exercise intervention. In contrast, the current study focuses on comparing the resting miRNA profiles of powerlifters and untrained individuals.

Comments 2: “7.           The miRNA profile after RE with protein consumption (Table 1? Are corresponding studies presented there? Please, reference the source). The authors note that this regime invokes Akt-mTOR channel with the consequent further responses.”

Response 2: Thank you for the comment! Part of the section not directly relevant has been moved to Section 6. In addition, we have supplemented this part with two chronic studies involving post-exercise protein intake, which are also included in Table 2.

Comments 3: “Ago2 network comprises a lot more proteins, e.g.: AGO1, AGO2, DDX20, DDX6, DHX9, DICER1, FMR1, GEMIN4, TARBP2, TNRC6A, TNRC6B. Please, ascertain if there are all of them in the EV.”

Response 3: Thank you very much for drawing our attention to this important detail! The incorporation of various proteins of the AGO network into extracellular EVs is indeed a significant aspect that may contribute to a deeper understanding of EV function. Accordingly, we have revised the “MiRNAs” section addressing this point, in the hope that it enhances the scientific value of the manuscript. For your reference, the newly added section can be found between lines 176 and 187.

Comments 4: “The authors state that “Different cell types (endothelial cells, myocytes, white blood cells, platelets, fat cells) can express different types of miRNAs (bound to proteins, packaged in EVs), which might influence the functioning of other cells, promoting cell-cell communication”. While the EV content may enter various cell types, it’s not clear if there are some specificity of cells target.”

Response 4: Thank you for your insightful comment regarding the specificity of extracellular vesicle (EV) targeting to recipient cells! We have addressed this important aspect by expanding Section 3 in the manuscript. Thus, although EV cargo can be delivered to multiple cell types, targeting mechanisms may confer a certain level of specificity, which we have now discussed in detail in the revised manuscript (Section 3). We hope this addition enhances the comprehensiveness of our study. The addition can be found between lines 125 and 136.

Comments 5: “Chapter “4. miRNAs.” could be renamed as “muscle specific miRNA’.”

Response 5: Thank you for your valuable suggestion! We have renamed the subtitles in Section 4 from “microRNA” to “miRNA,” while retaining the main section title as “miRNA.”, since muscle-specific miRNAs represent only a subtype and do not apply to all the miRNAs discussed in our review, particularly not to ci-miRNAs.

Comments 6: “76 : MiRNAs appearing in the blood circulation”

Response 6: Thank you for the suggestion! We have inserted the term ‘blood’ in line 213.

Comments 7: “Table 1 should be moved to Chapter 5.”

Response 7: Thank you for the comment. Indeed, Table 1 fits better within Section 5, and therefore we have moved it there.

Comments 8: “Chronic RE’ term should be discussed (elaborated) on the first instance.”

Response 8: Thank you for the helpful comment! At the first occurrence in the text (lines 77–78), we have clarified the meanings of both chronic and acute resistance exercise.

Comments 9: “The authors mention stress hormones involved in the physical training studies, which I think are the major players in the process. Notably, blood brain barrier doesn’t allow EVs to brain. Could the authors elaborate on this issue?”

Response 9: Thank you very much for this valuable comment! We fully agree that the hormonal system is a key factor facilitating adaptation. However, we found that extracellular vesicles (EVs), particularly small EVs, are capable of crossing the blood-brain barrier. Based on your insightful suggestion, we have supplemented the Introduction accordingly, highlighting the relevant section in yellow between lines 60 and 74.

Comments 10: “Conclusion section looks more like a discussion. It’s recommended to rename it, and provide a short conclusion section with major points.”

Response 10: Thank you very much for this helpful comment! We agree that the Conclusion section should be more concise; therefore, we have renamed the previous Conclusion as Discussion and subsequently added a brief, few-sentence Conclusion section.

Comments 11: “Please, check if all of abbreviations in the table are present/absent in the text; some are not comprehensive”

Response 11: Thank you very much for this helpful comment!
We carefully reviewed all abbreviations used throughout the manuscript and tables, and made the following changes to improve clarity and readability while avoiding unnecessary abbreviations:

  • IB Im: Likely remained in the table by mistake, so we have removed it.
  • EXPL: Appears in Table 2; for consistency and conciseness, we decided to keep it.
  • HI and LI: These abbreviations are used multiple times in Table 1 (and HI also in Table 2), so we retained them.
  • HIIE: Appeared in both Tables 1 and 2, but was removed for the sake of simplicity.
  • HR and LR: Each was mentioned only once across Tables 1 and 2, so we removed both abbreviations.
  • HYP: This abbreviation was replaced with MH, which had already been used multiple times in the text.
  • BMB: Since it only appeared in two instances, we decided to remove it.
  • CE, ddPCR, EE: Each of these was used only once or twice in the tables, so they were deleted.
  • MS, PLA, PRO: As these were mentioned only twice in the text or in Table 1, we removed them as well.
  • SE: Since it was used only once in the text, this abbreviation was also removed.

We believe these modifications improve the manuscript’s overall clarity. Thank you again for your careful and constructive suggestion!

Comments 12: “It's also unclear if all types of exersises mentioned are related to ER type, please, explain.”

Response 12: Thank you for your comment! The term resistance exercise (RE) is defined in the Introduction as a broad category of training involving muscle contractions performed against external resistance. This includes a wide range of exercise modalities, from hypertrophy-focused protocols to those targeting speed or strength endurance. At the same time, RE is clearly distinguishable from endurance or cardio-type training. Whenever the manuscript refers to a different type of exercise, it is explicitly specified in the text.

We would like to express our sincere gratitude for your thorough and detailed review of our manuscript! Your insightful comments and suggestions, both in terms of content and formatting, have significantly contributed to improving the clarity, structure, and overall scientific value of the paper. We believe that revisions made in response to your feedback have substantially strengthened the manuscript, and we are truly grateful for the time and effort you dedicated to its careful evaluation!

Round 2

Reviewer 1 Report

Comments and Suggestions for Authors

The manuscript has been significantly improved according to the reviewers' comments

Reviewer 2 Report

Comments and Suggestions for Authors

   The authors essentially addressed my comments, no further ones.